# FLIP-80M: 80 Million Visual-Linguistic Pairs for Facial Language-Image Pre-Training

## ABSTRACT

While significant progress has been made in multi-modal learning driven by large-scale image-text datasets, there is still a noticeable gap in the availability of such datasets within the facial domain. To facilitate and advance the field of facial representation learning, we present FLIP-80M, a large-scale visual-linguistic dataset comprising over 80 million face images paired with text descriptions. The construction of FLIP-80M utilizes large-scale publicly available image-text-pair dataset, filtering 5 billion samples from general domain, and incorporates with AI-Generated Content (AIGC) methods for quality management and data augmentation. The data creation process involves a mixed-method pipeline to filter face-related pairs from both visual and linguistic perspectives, including face detection, face caption classification, text de-noising, and AIGC augmentation. As a result, FLIP-80M stands as the largest face-text dataset to date. It shows exceptional data quality and demonstrates the potential to enhance the performance of face representation models. To assess the efficacy of our dataset, we use contrastive learning objective to train FLIP (Facial Language-Image Pretraining) and evaluate its representation capabilities across various downstream tasks. Experimental results reveal that our FLIP model achieves state-of-the-art results cross 10 different face analysis tasks like face parsing, face alignment, and face attribute classification. The dataset and models will be publicly available.

## CCS CONCEPTS

• **Computing methodologies** → **Computer vision**; *Language resources*; **Computer vision representations**.

## KEYWORDS

dataset, facial-linguistic, facial representation, CLIP model

**ACM Reference Format:**
Anonymous Author(s). 2018. FLIP-80M: 80 Million Visual-Linguistic Pairs for Facial Language-Image Pre-Training. In *Proceedings of Make sure to enter the correct conference title from your rights confirmation emai (Conference acronym 'XX)*. ACM, New York, NY, USA, 10 pages. https://doi.org/XXXXXXX.XXXXXXX

## 1 INTRODUCTION

In the rapidly evolving field of computer vision, the comprehension and interpretation of facial data are vital for various practical

applications. Recent years have witnessed significant progress in face analysis tasks, driven by deep neural networks with supervised learning. However, these supervised models are typically trained separately [6, 15–17, 52, 65] using manually annotated labels tailored for specific tasks, which imposes limitations on their capability to learn generalized facial representations.

In contrast to traditional manual labeling, recent efforts have explored an alternative approach to learning image representations directly from raw text data [20, 31, 44, 44] collected from the Internet. Notably, these studies have demonstrated that a simple pre-training task, involving the prediction of which caption corresponds to a given image, is an efficient way to learn generalized visual representations. As a result, this unique approach empowers the model for zero-shot transferability to various downstream tasks. In addition, a critical ingredient in these image-text models is the utilization of large-scale image-text data, necessitating millions of image-text pairs.

However, when it comes to the domain of facial analysis, the effectiveness of pre-training with natural language supervision is relatively unexplored, largely due to the absence of datasets designed specifically for this purpose. Although several pioneering efforts [3, 4] have been made to learn face representation from text descriptions, they still face some challenges. For instance, Talk2Face [33] aims to create a face-text dataset by converting image labels into textual descriptions, but these descriptions can only provide limited information. Similarly, LAION-Face [62], while promising, is built by filtering face-related pairs from a large openly available image-text dataset. However, it only focuses on detecting the presence of human face in images and does not check whether text is related to faces, which limits the relevance of the data from the perspective of the face domain.

To advance generalized facial representation learning with natural language guidance, we introduce FLIP-80M: a large-scale visual-linguistic dataset for Facial Language-Image Pretraining with over 80 million face-text pairs. Instead of collecting face images and texts from scratch, FLIP-80M is constructed based on the Large-scale Artificial Intelligence Open Network (LAION-5B) [49] and integrated with the latest AI-generated content (AIGC) models. To filter face-related pairs, we employ a mixture of automatic methods from both visual and linguistic perspectives, including face detection and face caption classification. Additionally, we incorporate large language model (LLM) to enhance the text descriptions and use large language-vision model (LVM) to augment the dataset with richer text captions and higher-resolution images. To the best of our knowledge, FLIP-80M stands as the largest face-text dataset to date.

To validate the value of FLIP-80M, we use contrastive learning objectives to train image and text representations initialized with CLIP weights, namely FLIP. We extensively evaluate it on 10 facial downstream tasks across different scenarios including face attribute

classification, face parsing, and face alignment. We measure the performance through zero-shot transfer and linear probing, with our results consistently surpassing other image-text baseline models (CLIP[44], FaRL[62], DataComp[14]). Moreover, our models also outperform previous task-specific supervised models. To further explore the impact of datasets on model performance, we also pretrain from scratch using different datasets including our FLIP-80M, LAION-Face[62] and CC12M[7]. On downstream tasks, the model trained by FLIP-80M can achieve better results, which suggests that our data has better quality (relevance and richness) in the face domain. The main contributions of this work can be summarized:

- We provide an extensive face-text dataset containing over 80 million paired instances, which is 3 times larger than currently available LAION-Face dataset. We also propose classification based text filtering and LLM based denoising to make the fine-grained linguistic description much more relevant with the paired face images, which offer a valuable resource for future research in face-related tasks.
- We propose a novel AIGC data construction pipeline, which serves as a high-quality data augmentation method, ablation study justifies the effectiveness of such augmentation.
- We validate our dataset's effectiveness by training the FLIP model and demonstrating its superiority over other image-text models, i.e. state-of-the-art performances in tasks such as face attribute classification, face parsing, and face alignment, are achieved. CLIP models trained from scratch using our dataset also shows better performance than that trained using other face-text datasets.

## 2 RELATED WORK

### 2.1 Face Datasets

**Conventional Datasets.** Traditional face analysis datasets are typically constructed with a task-centric approach, where face images are manually labeled with predefined task labels such as age, race, gender, expression, and other facial attributes [8, 22, 38, 41, 42]. We have summarized several datasets from different face analysis tasks in Table 1. For conventional approach, each dataset is customized to specific facial task, making them less suitable for training general facial representation models.

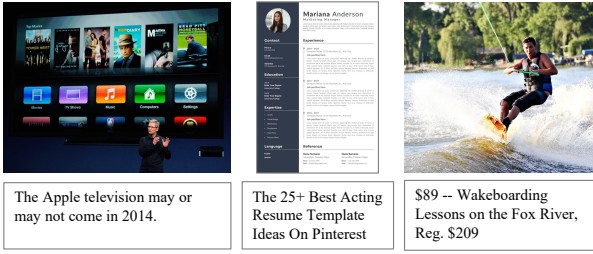

The Apple television may or may not come in 2014.

The 25+ Best Acting Resume Template Ideas On Pinterest

$89 -- Wakeboarding Lessons on the Fox River, Reg. $209

**Figure 1: Examples of ineffective face image and text pairs. Although faces appear in the images, the text descriptions are unrelated to the faces.**

**Face-Text Datasets.** To establish a connection between natural language and facial images, recent efforts, as summarized in Table

**Table 1: Existing face domain datasets. The upper section presents conventional labeled datasets, and features, while the lower section highlights face-text datasets.**

| Dataset | Samples | Supervision |
|---|---|---|
| CelebA [38] | 203k | 40 facial attributes |
| AffectNet [41] | 400k | 8 facial expressions |
| FS2k [12] | 2k | 24 features depict diverse scenes |
| ExpW [60] | 92k | 7 facial expressions |
| CACD [8] | 163k | age of 2,000 celebrities |
| IMDB-WIKI [45] | 523k | age and gender |
| FairFace [22] | 101k | race, gender, and age |
| FFHQ-Text [64] | 8k | manually annotated text |
| MM-CelebA-HQ [28] | 300k | text generated by syntax tree |
| CelebAText-HQ [51] | 150k | manually annotated text |
| Talk2Face [33] | 1M | collection of datasets |
| LAION-Face [62] | 20M | text-image pairs form Internet |
| FLIP (ours) | 83M | text-image pairs form Internet |

1, have introduced datasets consisting of images paired with corresponding textual descriptions [28, 33, 64]. However, these datasets face limitations in terms of diversity and scalability due to their rule-based design and heavy reliance on manual annotations. Closely related to our work is the recent LAION-Face dataset [62], which extracts a subset of 20 million samples containing face images from the LAION-400M dataset [50]. Nonetheless, as illustrated in Figure 1, LAION-Face concentrates detecting the presence of human face in images without taking text descriptions into account, leading to a large number of samples unrelated to the face domain. In this work, we adopt a mixed-method approach that filters face-related pairs from both visual and linguistic perspectives. This includes face detection, face caption classification, text de-noising, and AIGC augmentation. As a result,we are able to obtain data with stronger relevance and richness in the face domain, which has potential to promote face representation learning.

**Data Creation.** In computer vision community, constructing datasets often relies on manual annotation or supervised models to generate data labels. However, these approaches are limited by human labor costs and the model's inherent biases, leading to limitations in the quality and diversity of the resulting datasets. To address this issue, CLIP [44] utilizes naturally occurring image-text pairs collected from the Internet as supervisory signal. It greatly expanded the scale of the dataset and endowed the model with multimodal capabilities in both vision and language. Publicly available datasets like Laion-5B[49], CC12M[7] and DataComp[14] greatly stimulate research in this area. Nevertheless, due to the varying quality of these Internet-native image-text pairs, such data are mainly used for weakly supervised learning.

More recently, researchers have been leveraging large language models (LLMs) to construct datasets for instruction fine-tuning automatically. For example, Wang et al. [56] utilizes random seed topics to guide the model in generating question-answer pairs. In the multi-modal domain, ShareGPT4V [9] and LLAVA [35] employ vision-language models (VLMs) to annotate images. These approaches typically start with a fixed image dataset and use seed questions to generate image-text pairs with text conversation. However, self-instruct-based methods are often limited by the diversity

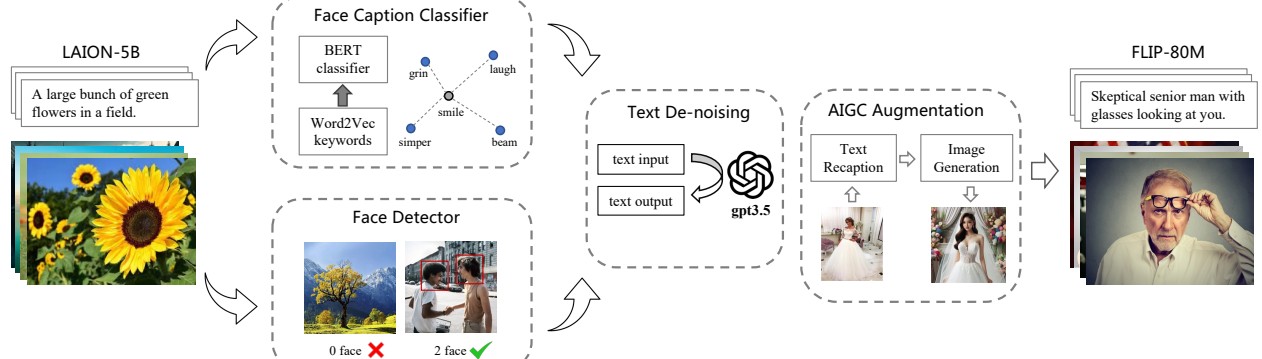

**Figure 2: Overview of the FLIP-80M dataset construction pipeline. The original data undergoes several processes, including face detection, face caption classification, text denoising, and AIGC augmentation, resulting in a total of 83 million face-text pairs (including 1 million AIGC samples).**

of the seed topics. Although the V/LLMs are able to generate different conversations, it is still constrained by the distribution of the original seeds. Moreover, previous methods primarily focus on generating (recaption) text based on given images, without redrawing the images. In this paper, we propose a AIGC-augmentation pipeline, which simultaneously generates both text and corresponding images while constraining the distribution using natural samples.

## 2.2 Facial Vision-Language Representation Learning

In order to learn more transferable visual models, recent models [18–20, 44] have made a large step forward in multimodal learning with large-scale image-text data. The core idea is to learn perception from supervision contained in natural language with contrastive objectives. After extensive pre-training, they are capable of associating visual concepts with natural language, facilitating zero-shot transfer to various downstream tasks. However, the specific impact of image-text pre-training in the face domain remains largely unexplored. While pioneering approaches have made attempts, there are still limitations. Talk2Face [33] introduces a general generative framework that unifies various tasks in a unified sequence-to-sequence format. However, it falls short in accurately representing faces, as texts are mainly converted from face labels. FaRL [62] attempts to learn facial representations by combining contrastive learning and masked image modeling, leveraging natural language supervision. In this work, we aim to explore the impact of data quality (face-domain relevance and richness) on face representation learning. We use vanilla CLIP framework to train FLIP model with our proposed dataset, the experimental results reveal its superiority in performance, highlighting the value of FLIP-80M in the face domain.

## 3 FLIP-80M DATASET

In this section, we present a comprehensive methodology for constructing our extensive and text-aligned human face dataset. Specifically, we filter face-related pairs from the LAION-5B dataset by

employing a combination of techniques from both visual and linguistic perspectives. In addition, we conduct a thorough analysis of the distribution and quality of our dataset, and compare it with existing related works. Notably, the meticulously curated FLIP-80M dataset represents the largest image-text dataset in the human face domain.

### 3.1 Construction Methodology

**Overview.** LAION-5B [49] is used as our data source, which is a publicly available dataset extracted from the Internet and filtered using the CLIP model. Our focus is specifically on English language text, which contains 2.3 billion samples. We design a pipeline to build our dataset from the raw data, involving the following steps:

**Visual Filtering for Face-Relevant Images.** To ensure the inclusion of visually relevant face images, we utilize the RetinaFace [10] detector to identify images containing human faces. Following LAION-Face, we selectively collect samples from LAION-5B with detection scores surpassing 0.9, resulting in approximately 200 million face-text pairs. This ensures a high-quality starting point for our dataset.

**Textual Filtering for Face-Relevant Descriptions.** We then train a text classifier to obtain textual descriptions associated with human faces. The initial step involves training a Word2Vec [40] model using text descriptions gathered by a face detector. We then specify 12 words that are strongly associated with human faces, i.e. *"smile", "nose", "eyes", "mouth", "cheek", "sad", "angry", "upset", "scared", "surprised", "eyeglasses", "earrings"*. By manual verification, texts containing these words have a higher probability of describing human faces. To broaden our exploration, we use the trained Word2Vec to find the top 10 nearest neighbors for each seed word, and subsequently extracted the 3 nearest neighbors for each of these resulting words, yielding a total of 360 words. After eliminating duplicates and conducting manual screening, we end up with a list of 155 words that are relevant to face descriptions. A word cloud of these keywords is shown in Figure 4.

Using these keywords, we recall the text in the pairs containing face images as positive samples, then randomly sample from the LAION-5B dataset as negative samples. We collect a total of 50,000

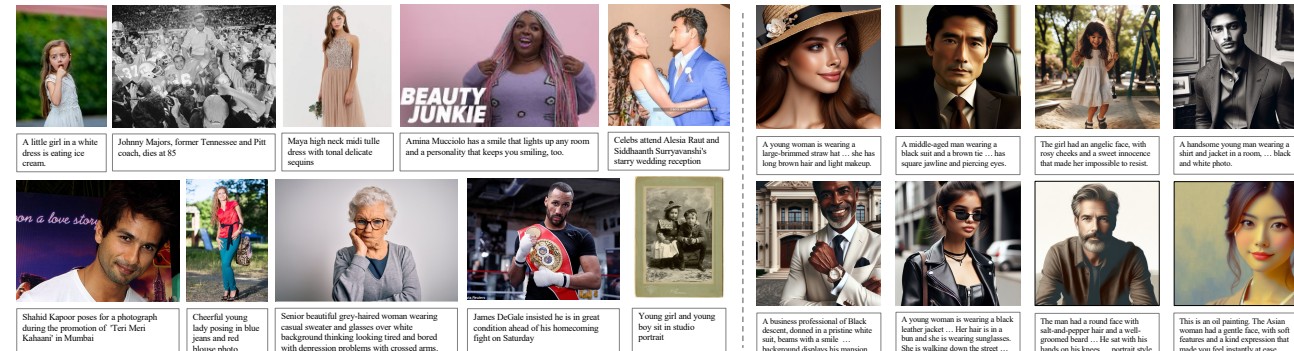

(a) Samples obtained from Internet | (b) Samples generated from AIGC augmentation

Figure 3: Data examples from FLIP-80M. (a) Samples obtained from Internet. (b) Samples generated from AIGC augmentation.

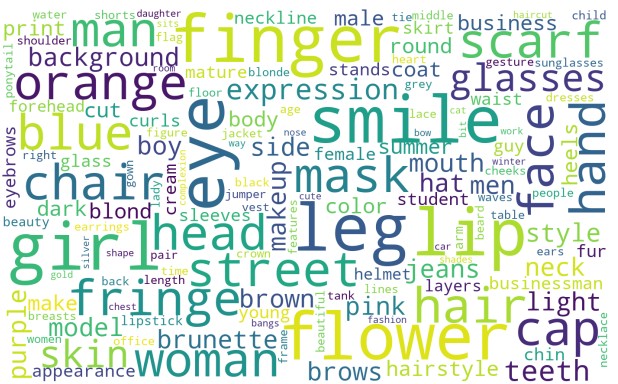

Figure 4: Word cloud of all keywords used for recalling positive samples.

samples, ensuring a 1:1 ratio of positive to negative samples. These samples are then used to fine-tune a BERT [24] text classification model. We use the classifier to filter the pairs obtained in the previous step. Any samples with text scores higher than 0.5 are included, resulting in a total of 82 million pairs.

**Text Denoising with Language Models.** Given that our data is sourced from the Internet through Common Crawl, it often includes irrelevant noise, such as gallery tags, HTML tags, escape symbols, and so on, which lacks semantic relevance. To address this, we employ a large language model (gpt3.5-turbo) for text denoising. Specifically, we instruct LLM with a combination of system prompt and few-shot examples. In the system prompt, we ask the language model to act as a text correction engine to optimize the quality of the input text. Followed by 3-shot examples and user input, the model is expected to directly output refined text. In this way, we systematically eliminate extraneous and noisy textual elements. This process significantly enhances the overall quality of our dataset.

**AI-Generated Content Augmentation.**

To further improve data quality, we design an AIGC-based data augmentation process aimed at constructing high-definition and richly described facial image-text samples. Firstly, given an image randomly sampled from our dataset, we utilize vision-language

model GPT-4 [1] to generate both general and face-specific captions. These captions are concatenated to form the recaptioned text description of the image. Then, we use text-to-image generation model DALL-E 3[2] to generate an $1024 \times 1024$ resolution image based on the above caption. The generated text and image are treated as an augmented sample. The process is illustrated in Figure 5.

Through this method, we produced 1 million text-image samples. Compared with the original samples, these samples feature richer text description and higher image definition, but more significantly, they show stronger image-text correlation. It serves as a high-quality subset of FLIP-80 which can used at a later stage of training to boost model performance.

Previous self-instruct AIGC data augmentation pipelines [36, 55] mainly use a list of seed topics or questions to generate new samples, which has limited diversity. In our method, we directly sample from naturally distributed images and use the AIGC tool to generate brand new samples that are completely different from the original samples, which enriches the diversity.

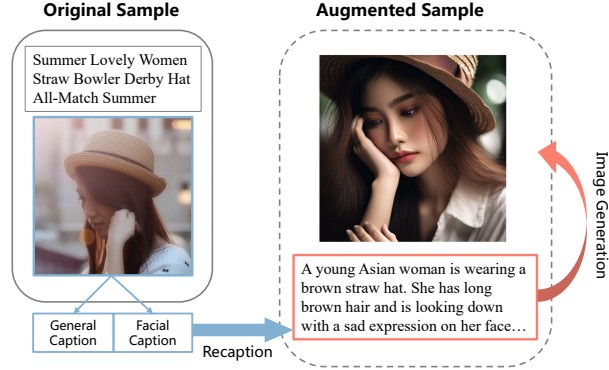

Figure 5: Illustration of the AIGC-based augmentation method. For an original sample, we first recaption the general and facial text descriptions, and then the descriptions are used to generate new image.

## 3.2 Data Analysis

**Quality.** We evaluate the quality of our data construction methodology from several perspectives. For the face detector, we employ the state-of-the-art RetinaFace method, which achieves an accuracy of 99.4% on the LFW dataset, which is sourced from real-world scenes, and the accuracy of human is only 97.5%. For the face caption classifier, we evaluate it on the test sets of MM-CelebA [28] and FFHQ-text [64], our classifier demonstrated an accuracy of 99.5% and 97.1%, respectively. For text denoising, we removed 9.5% of characters, and used the GPT2-large model to evaluate the text perplexity of the dataset. The results suggest that our denoising reduced the text perplexity from 88.5 to 68.6 on average (lower is better).

**Statistics.** In total, we collected 83 million text-image pairs. The mean length of text is 85.1 characters, 15.6 words. We perform statistics on the distribution of images, including age, race, and the number of faces contained in each image, and the results are shown in Figure 6. It can be seen that more than half of the faces are predicted to be between 20 and 29 years old. We speculate that this is because people in this age group are more active on the Internet and therefore provide more data. For race classification, white and black people account for more proportion. This is because only English texts are considered in the dataset, and English speakers from other races account for less. Some examples of FLIP-80M are shown in Figure 3.

**Data Release.** We release FLIP-80m dataset with two subsets. (1) 82 million text-image pairs crawled from the Internet, with different image resolutions. (2) 1 million of AIGC-based text-image pairs with a fixed $1024 \times 1024$ resolution. Metadata in the dataset contains the following fields: (1) The URL of the image. (2) The text description. (3) Keywords of text description, which can be used for clustering or retrieval. (4) Height and width of the image. (5) NSFW tag. Following LAION-5B, our data is released under CC-BY-4.0 license. For the AIGC-augmented subset, since the data is generated using commercial AIGC APIs, we are able to claim ownership of the data and release it under the CC-BY-4.0 license.

**Safety.** For the safety of dataset, we directly follow LAION-5B's pornographic and sexualized content classification (NSFW). 5.4% of images are detected as NSFW, which can be filtered out by a user with the NSFW tag.

## 4 EXPERIMENTS

In this section, we conduct experiments to evaluate the effectiveness of our proposed dataset. In the experiment, we use contrastive learning objectives [19] to train FLIP models initialized with CLIP weights. Then, we evaluate on 10 different downstream tasks covering face attribute classification, face alignment and face parsing. For ablation of training dataset, we also compare models pre-trained from scratch with the same hyperparameters using different data sources.

### 4.1 Training FLIP

**Training Objective.** We conduct facial language-image pre-training, denoted as FLIP, for generalized facial representation learning. Following CLIP [44], we adopt a contrastive objective to learn a similarity representation between text and face image within a batch of

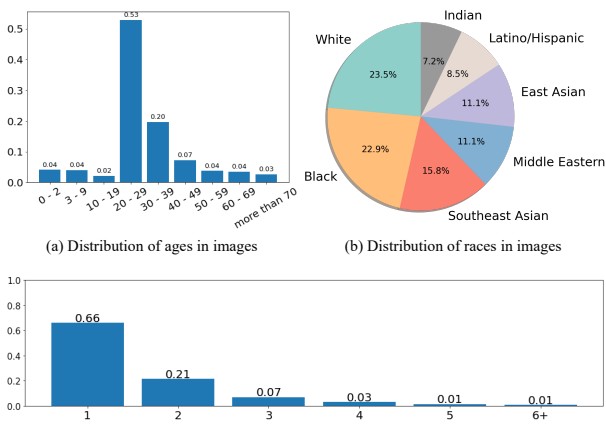

(a) Distribution of ages in images  (b) Distribution of races in images

(c) Distribution of the number of faces in images

**Figure 6: Overview of the statistics for our proposed dataset, detailing distributions across age, race, and the number of faces in images.**

$N$ pairs $\{T, I\}$. Specifically, the face images and texts are encoded using two transformers [11, 53] to extract feature representations, denoted as $F^T = \{f_1^T, f_2^T, ..., f_N^T\}$ and $F^I = \{f_1^I, f_2^I, ..., f_N^I\}$. The optimization objective aims to increase the cosine similarity of paired image and text features and decrease it for non-paired features. The loss function comprises the following two components:

$$L^I = -\frac{1}{N} \sum_{i=1}^{N} \log \frac{e^{\cos(f_i^I, f_i^T)}}{\sum_{j=1}^{N} e^{\cos(f_i^I, f_j^T)}} \quad (1)$$

$$L^T = -\frac{1}{N} \sum_{i=1}^{N} \log \frac{e^{\cos(f_i^T, f_i^I)}}{\sum_{j=1}^{N} e^{\cos(f_i^T, f_j^I)}} \quad (2)$$

**Architectures.** For fair comparison, we implement our model following prior works [44, 62]. Specifically, the text encoder is based on Transformer [53] and has an input length of 77 tokens. For the image encoder, we leverage different Vision-Transformer [11] models, including ViT-B/32, ViT-B/16, and ViT-L/14, each operating at a resolution of $224 \times 224$. Detailed model configurations are shown in Table 3

The models receive $224 \times 224$ resolution image as input and divide them into patches according to the frame size. For example, Vit-B/16 splits an image into $14 \times 14$ patches. In addition, a CLS token is placed before the input sequence as a global image representation, resulting in a total of 197 patches.

**Hyperparameters.** In the main experiment, our FLIP models use CLIP weights as initialization and post-train for 3 epochs with FLIP-80M. AdamW [39] with $\beta 1 = 0.9$, $\beta 2 = 0.999$ is employed for optimization. The learning rate is initialized with $2e^{-5}$ and batch sizes configured as 1,760 / 876 / 400. In the ablation study, we pre-train ViT-B/16 FLIP from scratch, using 10 million samples from different datasets and trained for 5 epochs. We use TencentPretrain [61] as our framework, and the pre-trained weights will also be released in the HuggingFace format.

**Table 2: Experiment results on face attribute classification tasks using linear probe and zero-shot evaluation.**

| Pre-training Settings | | Linear Probe | | Zero-Shot Performance | | | |
|---|---|---|---|---|---|---|---|
| | | | | Accuracy (%) ↑ | | | |
| Method | Architecture | FS2K | FairFace | RAF-DB | AffectNet | ExpW | CelebA |
| DataComp [14] | ViT-B/32 | 82.85 | 54.22 | 15.42 | 15.25 | 9.790 | 67.02 |
| | ViT-B/16 | 87.21 | 67.61 | 14.34 | 16.98 | 6.243 | 63.86 |
| CLIP [44] | ViT-B/32 | 87.69 | 76.72 | 29.01 | 27.70 | 16.59 | 71.15 |
| | ViT-B/16 | 87.02 | 77.76 | 26.86 | 31.85 | 12.30 | 67.38 |
| FaRL [62] | ViT-B/16@16 | 87.25 | 75.89 | 38.17 | 25.98 | 11.16 | 71.59 |
| | ViT-B/16@64 | 86.83 | 75.95 | 33.77 | 24.53 | 12.88 | 44.49 |
| FLIP (ours) | ViT-B/32 | 87.72 | 76.06 | 52.02 | 30.85 | 17.51 | 75.81 |
| | ViT-B/16 | 87.79 | 77.78 | 61.11 | 33.60 | 18.25 | **75.99** |
| | ViT-L/14 | **88.78** | **78.51** | **61.31** | **35.60** | **19.48** | 72.18 |

**Table 3: Configurations of FLIP models. V: vision; L: language.**

| Architecture | V/L-Layers | Hidden Size | V-patches | L-tokens |
|---|---|---|---|---|
| ViT-B/32 | 12 / 12 | 512 / 768 | 50 | 77 |
| ViT-B/16 | 12 / 12 | 512 / 768 | 197 | 77 |
| ViT-L/14 | 24 / 12 | 768 / 1024 | 257 | 77 |

## 4.2 Face Attribute Classification

**Dataset.** Face attribute classification aims to predict various attributes (like gender, age, race, and hair color) from a given face image. We assess our method's performance on six datasets, each offering unique challenges and characteristics. CelebA [38] is a large dataset with over 202K face images, each annotated with 40 attributes. FS2K [13] is a high-quality Facial Sketch Synthesis dataset, comprising 2,104 image-sketch pairs that span three sketch styles and encompass annotations for six facial attributes. FairFace [23], a race-balanced dataset, includes 108,501 face images annotated with information on race, gender, and age groups. RAF-DB [32], a large-scale facial expression database, offers around 30K diverse facial images annotated with seven basic emotions and 12 compound emotions. AffectNet [41], with approximately 0.4 million images, provides manual labels for eight facial expressions. Finally, ExpW [60] is a dataset tailored for facial expression recognition, featuring 91,793 faces labeled with seven basic expressions. This diverse collection of datasets allows for a comprehensive evaluation of our approach across various face attribute classification scenarios. Note that we adhere to the original train and test splits, with the exception of ExpW, where we utilize the entire dataset for zero-shot evaluation due to the absence of a predefined split.

**Experimental Setup.** Inspired by Ikezogwo et al. [19], Radford et al. [44], our experimental design incorporates two evaluation approaches. Firstly, we apply linear probes to assess the model's performance on the FS2K and FairFace datasets. Secondly, we conduct zero-shot learning experiments on the remaining datasets. For linear probes, we extract image features from the vision transformer of our FLIP and subsequently train a single linear classifier for attribute classification, following the methodology outlined in CLIP [44]. For zero-shot learning, we calculate the cosine similarity between image embeddings and possible text embeddings, then obtain the probability distribution via a softmax. Specific prompt

templates such as "a photo of a face with {label} expression" are utilized to extract text embedding for RAF-DB, AffectNet, and ExpW dataset. In the case of CelebA, we employ "a photo of a face with {attribute}" and "a photo of a face without {attribute}" as the templates. Our evaluation is attribute-specific and we present the mean accuracy for each dataset. In addition, all the images are resized to $224 \times 224$ in our experiments.

**Results.** The outcomes are displayed in Table 2. As we can see, our models achieve superior performance. In the comparison of models with the same architecture (e.g. ViT-B/16), our model shows better performance. In addition, our models perform noticeably better on zero-shot setting, which mainly benefit from the fine-grained description of face attributes in our dataset. These results demonstrate the effectiveness of our dataset in enhancing facial feature learning, positioning it as a valuable resource for advancing facial attribute classification.

## 4.3 Face Alignment

**Dataset.** Face alignment is a task to regress 2D face landmark coordinates in a given face image. We leverage two widely used datasets for evaluation: AFLW-19 [66] contains 20,000 training images and 4,386 testing images, annotated with 19 landmarks; 300W [46–48] includes 3,837 training images and 600 testing images, each annotated with 68 landmarks.

**Experimental Setup.** In line with FaRL [62], we train a head on top of our FLIP model to achieve face alignment. To assess the few-shot and full-shot performance of various models, we conduct training with 1%, 10%, and 100% of the training data. The data split also follows FaRL's setting. We represent groundtruth landmark points as Gaussian heatmaps with a size of $128 \times 128$, normalized to a range of 0 to 1. The head is trained with a soft-label cross-entropy loss, and UperNet [59] is utilized to output the heatmap logits. Evaluation is conducted using the normalized mean error (NME) as the metric.

**Results.** The results, as presented in Table 4, consistently highlight the superiority of our model, $FLIP^{L/14}$, across different portions of the training data. Additionally, our approach outperforms models specifically designed for face alignment tasks on both AFLW-19 and the 300W dataset, as demonstrated in Table 5 and 6.

**Table 4: Face alignment results on AFLW-19 and 300W datasets, evaluated using NME$inter-diag$ ↓ and NME$inter-ocular$ ↓ as metrics, respectively. ↓ means lower is better.**

| Method | AFLW-19 | | | 300W | | |
|---|---|---|---|---|---|---|
| | 1% | 10% | 100% | 1% | 10% | 100% |
| DataComp$^{B/16}$[14] | 1.40 | 1.15 | 1.01 | 4.74 | 3.51 | 3.10 |
| CLIP$^{B/16}$[44] | 1.30 | 1.11 | 0.995 | 4.18 | 3.42 | 3.08 |
| FaRL$^{B/16}$[62] | 1.35 | 1.15 | 0.991 | 4.25 | 3.42 | 3.12 |
| FLIP$^{B/32}$ | 1.38 | 1.18 | 1.04 | 4.79 | 3.67 | 3.29 |
| FLIP$^{B/16}$ | 1.30 | 1.12 | 0.987 | 4.27 | 3.41 | 3.07 |
| FLIP$^{L/14}$ | **1.25** | **1.09** | **0.973** | **4.02** | **3.29** | **2.99** |

**Table 5: Comparison with state-of-the-art face alignment methods on two AFLW-19 test sets: Full set and Frontal subset.**

| Method | NME$_{inter-diag}$ ↓ | | NME$_{box}$ ↓ | AUC$^7_{box}$ ↑ |
|---|---|---|---|---|
| | Full | Frontal | Full | Full |
| ATF [27] | 1.55 | - | - | - |
| LUVLi [25] | 1.39 | 1.19 | 2.28 | 68.0 |
| MHHN [54] | 1.38 | 1.19 | - | - |
| DTLD+ [29] | 1.37 | - | - | - |
| SHR-FAN [5] | 1.31 | 1.12 | 2.14 | 70.0 |
| LAB (w/ B) [57] | 1.25 | 1.14 | - | - |
| DataComp [14] | 1.01 | 0.876 | 1.432 | 80.0 |
| FaRL [62] | 0.991 | 0.851 | 1.402 | 80.4 |
| FLIP$^{B/32}$ | 1.04 | 0.894 | 1.47 | 79.5 |
| FLIP$^{B/16}$ | 0.987 | 0.854 | 1.396 | 80.4 |
| FLIP$^{L/14}$ | **0.973** | **0.842** | **1.376** | **80.7** |

**Table 6: Comparison with state-of-the-art face alignment methods on three 300W test sets: Common subset, Challenge subset, and Full set.**

| Method | NME$_{inter-ocular}$ ↓ | | |
|---|---|---|---|
| | Common | Challenge | Full |
| PIPNet [21] | 2.78 | 4.89 | 3.19 |
| SLPT [58] | 2.75 | 4.90 | 3.17 |
| FaRL [62] | 2.69 | 4.85 | 3.12 |
| HIH [26] | 2.65 | 4.89 | 3.09 |
| DataComp [14] | 2.70 | 4.75 | 3.10 |
| CLIP [44] | 2.69 | 4.68 | 3.08 |
| RePFormer [30] | - | - | 3.01 |
| SPIGA [43] | **2.59** | 4.66 | **2.99** |
| FLIP$^{B/32}$ | 2.89 | 4.91 | 3.29 |
| FLIP$^{B/16}$ | 2.68 | 4.66 | 3.07 |
| FLIP$^{L/14}$ | 2.62 | **4.51** | 2.99 |

### 4.4 Face Parsing

**Dataset.** Face parsing is a task to predict per-pixel labeling of face images. We utilize two widely used datasets for this task: LaPa [37] and CelebAMask-HQ [28]. LaPa comprises over 22,000 images, with 18,176 designated for training and 2,000 for test. Each image is annotated with 11-category pixel-level labels. CelebAMask-HQ consists of around 30,000 facial images, with 24,183 allocated for training

**Table 7: Face parsing results on CelebAMask-HQ and LaPa, with F1 scores (%) ↑ as evaluation metrics. ↑ means higher is better.**

| Method | CelebAMask-HQ | | | LaPa | | |
|---|---|---|---|---|---|---|
| | 1% | 10% | 100% | 1% | 10% | 100% |
| DataComp$^{B/16}$[14] | 81.18 | 85.47 | 87.54 | 87.79 | 90.66 | 91.91 |
| CLIP$^{B/16}$[44] | 82.18 | 85.73 | 87.75 | 88.13 | 90.91 | 92.21 |
| FaRL$^{B/16}$[62] | 81.50 | 85.10 | 86.72 | 88.21 | 90.91 | 92.32 |
| FLIP$^{B/32}$ | 77.52 | 81.78 | 84.38 | 83.77 | 87.23 | 88.48 |
| FLIP$^{B/16}$ | 82.25 | 85.74 | 87.80 | 88.47 | 90.94 | 92.18 |
| FLIP$^{L/14}$ | **83.32** | **86.80** | **88.41** | **89.11** | **91.44** | **92.46** |

and 2,824 for test. Each image in CelebAMask-HQ is annotated with a 19-category label map.

**Experimental Setup.** Similar to the FaRL's training and test settings [62], we train a head on top of our FLIP model to achieve face parsing. The non-cls tokens on each layer of FLIP are reshaped into a 2D feature map ($14 \times 14$) and transformed into multiple feature maps using UperNet [59] and a $1 \times 1$ convolution. AdamW with a learning rate 1e-3 and weight decay $1e^{-5}$ are used for the optimization. We also use Tanh-warping [37] to balance the performance across inner facial components and the hair region. F1 scores of facial components are used to measure the performance [37, 52].

**Results.** We also evaluate the few-shot and full-shot performance of various models by conducting training with 1%, 10%, and 100% of the training data. The results are shown in Table 7. Our larger model, FLIP$^{L/14}$, consistently outperforms CLIP, DataComp, and FaRL by a considerable margin. Moreover, our smaller model, FLIP$^{B/16}$, demonstrates superior performance in most cases. These results suggest that the model's effectiveness extends across various training scenarios, showcasing its capacity to learn robust and generalized facial features.

### 4.5 Comparison of Visualized CAMs

We conduct a comprehensive comparison of the visualized Class Activation Maps (CAMs) from FLIP, FaRL, and CLIP, employing the CLIP-ES framework [34], based on GradCAM [63]. The feature maps are corresponding to the layer preceding the final self-attention layer in the Vision-Transformer. Figure 7 presents the generated CAMs from different models. Notably, FLIP demonstrates superior accuracy in localizing facial areas through text queries. For instance, in the context of hair, FLIP's CAM comprehensively covers the entire hair region, outperforming other models that only capture a portion. Similarly, in the example involving earrings, CLIP activates the entire ear, while FLIP precisely identifies the position of the earring. These results demonstrate the potential of our model to learn more nuanced and detailed facial features.

### 4.6 Ablation Study

In previous experiments, we have compared the performance of the FLIP model. However, since pre-training involves complex processes and huge consumption of computing resources, it is difficult for us to reproduce all models from scratch under the same circumstances. To eliminate the impact of different pre-training settings

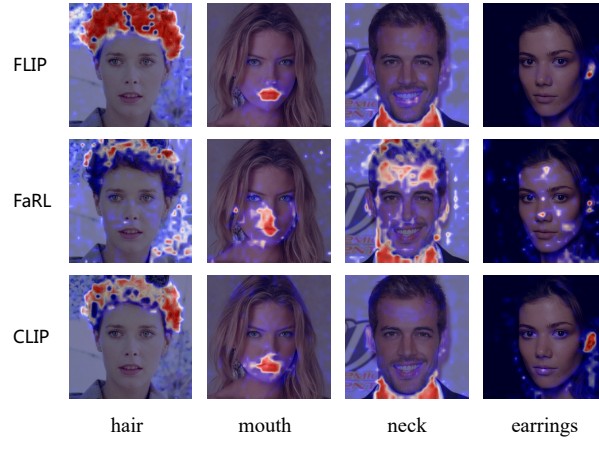

hair          mouth          neck          earrings

**Figure 7: Visualization of Class Activation Maps (CAM) generated with different text queries.**

**Table 8: Ablation experiment using different datasets to pretrain image and text contrastive models from scratch. Aug: AIGC Augmented subset.**

| Dataset | RAF-DB Accuracy (%) ↑ | CelebAMask-HQ F1 scores (%) ↑ | 300W NME$_{inter-ocular}$ ↓ |
|---|---|---|---|
| CC12M | 14.77 | 80.54 | 3.60 |
| LAION-Face | 23.04 | 84.85 | 3.34 |
| FLIP (w/o Aug.) | 25.91 | 85.34 | **3.26** |
| FLIP (w/ Aug.) | **29.11** | **85.52** | **3.26** |

(for example, FaRL uses additional training targets, CLIP uses larger-scale distributed training), we set up ablation experiments to train the models from scratch. It seeks to evaluate the datasets' influence on model performance in more detail.

Specifically, we use 10M data sampled from different datasets for training, including general dataset CC12M [7], face domain dataset LAION-Face [62], our FLIP-80M w/o AIGC augmentation and FLIP-80M. The models are randomly initialized and pre-trained for 5 epochs and then evaluated on representative downstream tasks.

The results are shown in Table 8. It can be seen that among models trained with a consistent amount of data, FLIP-80M can enable the model to achieve better results on downstream tasks. Moreover, we discover that adding AIGC-enhanced data produces higher results.

To further explore the impact of dataset quality (image-text relevance and richness) on model performance, we conduct an evaluation around data quality. We adopt both human and automated evaluation. In human evaluation, we divide the text richness and image-text relevance into 5 levels from the face domain aspect. The evaluation criteria and details are shown in the Appendix. For automatic evaluation, we use CLIP score and text perplexity (PPL), which are frequently used evaluation metrics of image-text relevance and text quality.

It can be seen from the results in Table 9 that our proposed FLIP-80M surpasses the previous related work LAION-Face in terms of data quality. In addition, the data produced using AIGC-augmentation obtains much better scores, which indicates that data

**Table 9: Evaluation result of image-and-text relevance and richness in each dataset. Aug: AIGC Augmentation subset.**

| Dataset | Human Eval. | | Automatic Eval. | |
|---|---|---|---|---|
| | Richness↑ | Relevance↑ | CLIP score↑ | Text PPL↓ |
| LAION-Face | 2.4 | 3.0 | 14.53 | 115.19 |
| FLIP(w/o Aug.) | 2.9 | 3.2 | 15.50 | 68.62 |
| AIGC-Aug. | **4.3** | **4.1** | **17.02** | **6.36** |

with high correlation and richness can boost the performance of face representation learning.

## 5 DISCUSSION

**Comparison with FaRL.** The most related work FaRL[62] primarily focuses on model structure and training methodology, it jointly learns from contrastive learning and masked image modeling. In this work, we focus on the importance of data quality. By creating better quality data, we are able to attain better results while using the vanilla CLIP training framework. Training data and methodology are two key factors that support face representative learning, FLIP and FaRL concentrate on these two aspects respectively. We hypothesize that further improvements can be achieved by combining FaRL's training method and FLIP-80M dataset, which we leave for future work.

**The benefits of FLIP-80M.** Inspired by the scaling law and the success of large language models, the scaling of model parameters and data size has gained popularity recently in multi-modal and computer vision research. These works are supported by large-scale datasets, and FLIP-80M fills the gap of data resource in the face domain. Our experiments are limited to face representation learning due to computational resource constraints, but FLIP-80M can be applied to a wider range of scenarios. Specifically, high-quality samples produced by AIGC-augmentation can be utilized for training face generation, editing, and question-and-answer models.

**Limitations.** Despite the value of the FLIP-80M dataset for facial domain research, there are two limitations that need to be considered. First, FLIP-80M is constructed upon the LAION-5B dataset, introducing potential biases and imbalances in data distribution. To mitigate this, we employ a synthesis-based augmentation method. However, the generated samples constitute only about 1.2% of the entire dataset, limiting their impact on the overall distribution. Additionally, the performance of this augmentation method remains underexplored due to a lack of comprehensive evaluation methods.

Second, our data processing pipeline incorporates multiple models, introducing the potential for cumulative errors that may impact overall data quality. Manual evaluation of the dataset reveals approximately 6% of samples with false positives, resulting from errors in the face detection or text classification models.

## 6 CONCLUSION

This paper presents FLIP-80M, a large-scale visual-linguistic dataset containing over 80 million face-text pairs. In experiments, we fine-tune the CLIP model using FLIP-80M, referred to as FLIP. The performance of FLIP is evaluated in various downstream tasks, highlighting the impact of data quality on face representation. Overall, this work contributes a valuable data source for future research. Datasets and pre-trained models will be publicly available.

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
