# OpenReview forum: "FLIP-80M: 80 Million Visual-Linguistic Pairs for Facial Language-Image Pre-Training"
_acmmm.org/ACMMM/2024/Conference — MM2024 Oral_

### Official Review · Reviewer_Md5u · 2024-05-16

**Rating:** 5
**Confidence:** 3

**Summary:**

This paper constructed a dataset FLIP-80M using AIGC pipeline with LLM denoising, and used contrastive learning to pre-training ViT architectures to evaluate the performance.

**Strengths:**

1. The constructed dataset greatly contributes to the facial representation learning domain.
2. The construction pipeline is well-organized.
3. The analysis of the dataset quality is in-depth.
4. The experiments show great performance.

**Limitations:**

1. Have you considered making comparisons or discussing with other large-scale single-modal face datasets, such as MS-Celeb-1M and Glint360K?

**Suitability:**

3

---

### Official Review · Reviewer_ociM · 2024-05-21

**Rating:** 4
**Confidence:** 3

**Summary:**

This paper constructs a large face dataset. First, 82 million image-text pairs were obtained from the LAION-5B dataset using face detection and face text classification methods. Then, GPT-3.5 Turbo was used for denoising the image-text annotations. Finally, the DALL-E 3 model was used to generate an additional 1 million images, forming the FLIP-80M containing 83 million image-text pairs. The model trained on FLIP-80M achieved good results in three task scenarios: face attribute classification, face alignment, and face parsing.

**Strengths:**

(1) I believe the open-sourcing of the FLIP-80M dataset will have significant meaning for some related research fields.

(2) The amount of work involved of this paper is substantial. This paper has created the largest face dataset to date, with over 80 million image-text pairs, which is 3 times larger than the LAION-Face dataset. Additionally, comparative experiments in three scenarios, including face attribute classification, face alignment, and face parsing, were conducted, validating the effectiveness of this dataset.

(3) The paper is well-written and easy to understand.

**Limitations:**

(1) The novelty is a bit limited. The use of the RetinaFace detector, the BERT text classification, GPT-3.5 Turbo for text denoising, and DALL-E 3 for text generation are all common methods.

(2) In the AIGC image generation process, GPT-4 is used to generate both general and face-specific captions for the input images, which are directly concatenated. Doesn't this cause semantic redundancy?

(3) Table 8 and 9 show that the quality of the 1 million image-text pairs generated by DALL-E 3 is good, but they only account for a small portion of FLIP-80M. The author could consider using the open-source model Stable Diffusion XL (SDXL) to obtain more generated images.

(4) The authors consider the process of AIGC image generation as one of the contributions of this paper in line 135, but this seems insufficient. Comparing the generated images from DALL-E 3 and SDXL could better highlight this contribution.

**Suitability:**

3

---

### Official Review · Reviewer_Us7p · 2024-05-22

**Rating:** 5
**Confidence:** 3

**Summary:**

This paper constructs a large-scale image-text dataset containing more than 80 million face images and related text descriptions by filtered and augmented large-scale Internet data with mixed-method pipeline and AIGC technique. This paper illustrates that FLIP model pre-trained on this dataset performs better on multiple face analysis tasks, and has higher quality and data relevance than existing datasets.

**Strengths:**

1. The proposed dataset is with high quality and large scale. The construction of the dataset is achieved by filtering from large-scale Internet data and combining AIGC technology to improve the correlation of vision-language data through face detection, facial description classification, and text denoising.
2. The test indicators of the experiment part are diverse and the verification process is sufficient. It was verified from multiple perspectives that the model pre-trained on this dataset performed well on downstream tasks, including face attribute classification, face resolution, and face alignment, better than other image-text models and task-specific supervised models.
3. The text description of the dataset is of high quality and the image data is of high resolution. In the process of data set construction, AIGC technology is used to augment the data, and the text data is denoised by powerful pre-trained language model, which enriches the diversity of the dataset and improves the quality of it.

**Limitations:**

1. I am not sure whether the number of image-text pairs generated by AIGC, 1 million, is reasonable or not. In other words, it is not explained why there are 1 million image-text pairs generated by AIGC technology, instead of more or less.
2. The name FLIP gives me the feeling that this is an innovative algorithm structure because it is given a new name. However, according to the description of the paper, it was only pre-trained using FLIP-80M on the structure of CLIP. I think the current naming of FLIP may not be appropriate and may make readers confused. It would be better if the name could be changed to make the relationship between it and CLIP clearer.

**Suitability:**

3

---

### Official Review · Reviewer_qibV · 2024-05-24

**Rating:** 4
**Confidence:** 4

**Summary:**

The paper presents an 80M face-text dataset, FLIP. The construction pipeline includes visual-linguistic filtering and AIGC augmentation. The dataset is tested across 10 different face analysis tasks.

**Strengths:**

The dataset is large-scale and novel, and the experiments are extensive.

**Limitations:**

### 1. The main concern is whether the dataset benefits the downstream tasks.
a. I can see using FLIP improves representations in some settings. However, there is a gap with SOTA, e.g., the best accuracy of RAF-DB, CelebAMask-HQ, and AFLW-19 is 92.57%, 89.56%, and 0.92% (from paperwithcode.com).

b. Can you show the distribution of facial expressions and facial attributes? From the data examples, I feel the diversity of facial expressions is limited.

### 2. There are some places that make me confused.
a. Table 1 shows that LAION-Face and FLIP have the same supervision of text-image pairs from the Internet. Do you mean their difference is the sample quantity?

b. I am confused about what positive and negative samples are in L#345 - L#347

c. Why do the keywords contain 'print' in Figure 4?

d. How can the text classifier filter the pairs in L#386?

**Suitability:**

3

---

### Meta-Review · Area_Chair_UfuN · 2024-07-15

**Recommendation:** Accept (Oral)
**Confidence:** 5

**Metareview:**

All four reviewer impressions are in favor of the paper, and therefore I recommend acceptance. Authors are nevertheless required to address reviewer concerns in the camera ready version.